# EARLY WEIGHT AVERAGING MEETS HIGH LEARNING RATES FOR LLM PRE-TRAINING

## ABSTRACT

Training Large Language Models (LLMs) incurs significant cost; hence, any strategy that accelerates model convergence is helpful. In this paper, we investigate the ability of a simple idea – checkpoint averaging along the trajectory of a training run – to improve both convergence and generalization quite early during training. Here we show that models trained with high learning rates observe higher gains due to checkpoint averaging. Furthermore, these gains are amplified when checkpoints are sampled with considerable spacing in training steps. Our training recipe outperforms conventional training and popular checkpoint averaging baselines such as exponential moving average (EMA) and stochastic moving average (SWA). We evaluate our training recipe by pre-training LLMs, where high learning rates are inherently preferred due to extremely large batch sizes. Specifically, we pre-trained nanoGPT-2 models of varying sizes—small (125M), medium (335M), and large (770M)—on the OpenWebText dataset, comprised of 9B tokens. Additionally, we present results for publicly available Pythia LLMs, ranging from 1B to 12B, which were trained on the PILE-deduped dataset containing 207B tokens. Our code will be available on Github.

## 1 INTRODUCTION

Large Language Models (LLMs) have made a significant leap from billion to trillion scale, both in terms of parameters (Chowdhery et al., 2022; Ren et al., 2023) and pre-training data size (Hoffmann et al., 2022) Touvron et al. (2023a;b). This surge in both data and model size has rendered LLM pre-training increasingly slow and resource-intensive. For instance, a Llama 2 70B model trained with 2T tokens took 1720K GPU hours to train. To accelerate the training process, it is a popular practice in LLM pre-training (Biderman et al., 2023; Touvron et al., 2023a) to utilize exceptionally large batch sizes, thereby ensuring maximal GPU utilization and consequently necessitating the use of relatively high learning rates (Goyal et al., 2017).

In this paper, our goal is to improve the test generalization (log perplexity) of LLM pre-training while reducing the number of training steps, all without increasing the compute budget. To achieve this, we first demonstrate that: (a) models trained with higher learning rates exhibit greater improvements when averaged along the training trajectory, and (b) averaging distant model weights from a single training trajectory further amplifies these gains. We integrate these two insights to adapt LAWA (LAtest Weight Averaging) (Kaddour, 2022)—a technique that performs checkpoint averaging throughout a training trajectory using a sliding window—for pre-training LLMs.

We evaluate our methodology by pre-training nanoGPT-2 models of various scales, specifically 125M (small), 355M (medium), and 770M (large), using the OpenWebText dataset, which comprises 9B tokens. The experiments with nanoGPT-2 are conducted in a controlled environment to gain a deeper understanding of our training recipe. Furthermore, we extend our evaluation to publicly available Pythia LLMs (Biderman et al., 2023), which include model sizes of 1B, 2.8B, 6.9B, and 12B, trained using 207B tokens. Our experiments with Pythia LLMs aim to demonstrate the impact of our work on real-world LLMs.

**Main Contributions.** In summary, our findings are as follows,

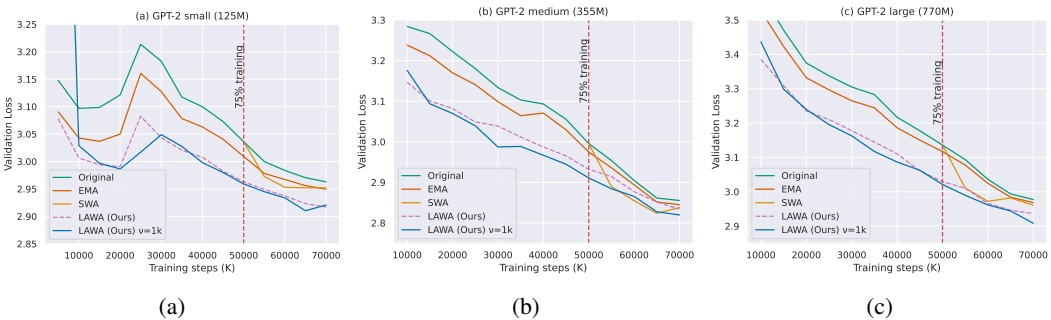

Figure 1: Across all model sizes, LAWA achieves faster convergence and generalizes better in comparison to original pretraining run and other baseline averaging schemes. Validation loss on Open-WebText with 70K training steps; (a) GPT2-small (125M) with Original is 2.963, EMA is 2.949, SWA is 2.952 and LAWA (ours-best) is 2.917, (b) GPT2-medium (355M) with Original is 2.855, EMA is 2.845, SWA is 2.837 and LAWA (ours-best) is 2.819, and (c) GPT2-large (770M) with Original is 2.977, EMA is 2.968, SWA is 2.961 and LAWA (ours-best) is 2.908.

1. We empirically show that models trained with high learning rate (LR) show pronounced gains over original training on performing checkpoint averaging very early on during training (Figure 2). This gain further amplifies when we sample distant checkpoints in the training run (Figure 10). We provide a intuitive explanation of this phenomenon in Section 2.

2. We observe that the training trajectory of LAWA closely mimics that of a model being trained with a low LR. The primary advantage of LAWA is that it allows LLMs to be trained with high LR without compromising generalization (Section 4).

3. We show that LAWA improves test generalization with fewer training steps compared to original training starting very early on during training; for both nanoGPT-2 and Pythia LLMs (Figure 1 and Figures 4-8). For Pythia checkpoints we perform zero-shot evaluation and observed similar gains, as shown in Table 2.

4. We compare our recipe with conventional training and popular baselines such as Exponential Moving Average - EMA (Szegedy et al., 2015) and Stochastic Weight Averaging - SWA (Izmailov et al., 2018). These baselines were not originally proposed or evaluated for LLM pre-training, but we adapt them to set meaningful baselines. Our training recipe outperforms conventional pre-training, EMA, and SWA.

5. Additionally, we perform a preliminary investigation of early weight averaging for a diffusion model for image generation (specifically, a 422M sized UNet model trained with the standard DDPM objective (Ho et al., 2020)). We observe thematically similar improvements (evaluated by the FID metric) as shown in Figure 9.

**Paper outline.** The remainder of this paper is structured as follows: We start by explaining the problem through a toy example and intuition behind our approach in Section 2. Following that we detail our experimental set up in Section 3 and present our main results in Section 4. Thereafter we discuss our work relative to prior works in Section 5. Finally, we conclude the paper with a summary of our conclusions and potential directions for future research.

## 2 INTUITION AND METHOD

### 2.1 TOY SETTING

We explain the setting using a simple toy problem of minimizing a two-dimensional loss function, represented as $L(w_1, w_2)$, where $w_1$ and $w_2$ are the parameters of the model. In this scenario, there exists an optimal batch size, $\mathcal{B}_0$, and an optimal learning rate, $\eta_0$, that minimizes the loss function. Assuming, for reasons outlined in Section 1, we are compelled to use a batch size, $\mathcal{B}$, and a learning

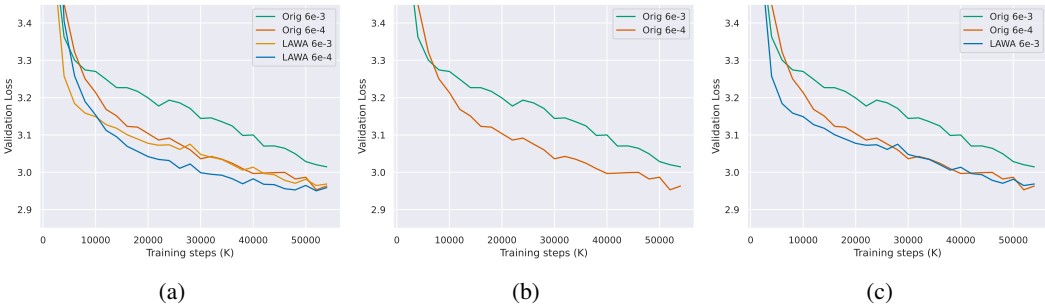

|   |   |   |
|---|---|---|
| (a) | (b) | (c) |

Figure 2: We compare two independently trained nanoGPT-2 (125M) models with LR = $[6 \times 10^{-3}, 6 \times 10^{-4}]$ on OpenWebText data. (a) Pre-training curve with and without LAWA. LLMs trained with higher LR observes higher gain due to LAWA. (b) The model trained with a high LR generalizes poorly compared to its counterpart trained with low/tuned LR. (c) The generalization gap caused by the high LR is effectively mitigated by LAWA.

rate, $\eta$, such that both are significantly larger than their optimal counterparts, i.e., $\mathcal{B} \gg \mathcal{B}_0$ and $\eta \gg \eta_0$ respectively. Suppose also that the loss function $L(w_1, w_2)$ exhibits a much higher curvature along $w_1$, as compared to $w_2$. It is widely known that the updates from the AdamW optimizer are mostly uniform across all weight dimensions (Liu et al., 2023). When $\eta \gg \eta_0$ the weight updates of $w_2$ will be accelerated, however this will cause oscillations along $w_1$ which in a long run adversely affects the convergence of $w_1$. A naive approach to mitigate this problem is to use a smaller LR or decay LR to 0, which might hinder progress in flatter regions. It is conceivable that a LLM might exhibit extremely heterogeneous curvatures exacerbating this issue during pre-training.

## 2.2 INTUITION OF OUR APPROACH

**Optimization Viewpoint.** We propose performing checkpoint averaging of model weights relatively early during training with high learning rates ($\eta$). The rationale behind this step stems from the fact that checkpoint averaging serves as a surrogate to LR decay, as demonstrated by Sandler et al. (2023). However, this surrogate LR decay is decoupled from the weight update during optimization, as checkpoint averaging is conducted in a post-hoc manner. Employing this simple technique, we mitigate the oscillations in $w_1$ while swiftly traversing through $w_2$, achieving enhanced generalization in fewer training steps as illustrated in Figure 3.

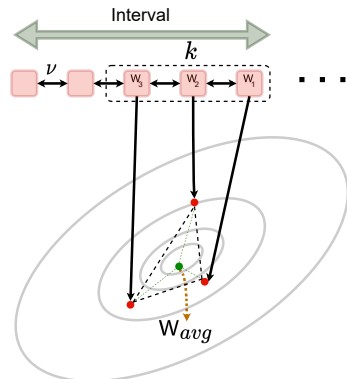

Figure 3: LAWA illustration: Given weights $W_1, W_2, ...W_k$ from a high LR trajectory separated by k-stepsize ($\nu$, Algorithm 1), LAWA computes $W_{avg}$ at a given step.[1]

**Diversity Viewpoint.** The practice of averaging the weights of model checkpoints is broadly recognized as being functionally analogous to ensembling (Izmailov et al., 2018; Wortsman et al., 2022). In model ensembling literature it is well established that diverse models improve the performance of the ensemble (Lakshminarayanan et al., 2016). Therefore, it is fair to assume that this principle also applies to model averaging as well. In our context, we define the diversity of a model at two distinct training steps, $\frac{1}{N} \sum_{i=1}^{N} \mathbf{1} \left[ y_i^{W_1} \neq y_i^{W_2} \right]$ which calculates the number of disagreements between the two checkpoints. This equation computes the number of samples from the same held-out set where the check-

---

[1]Note that W refers to the set of weights $\{w_1, w_2, ...\}$.

point $W_1$ disagrees with checkpoint $W_2$. A recent study by Athiwaratkun et al. has demonstrated that a higher LR can result in the generation of diverse model checkpoints. We observed (Figure 1) that this phenomenon can be further amplified by sampling far apart checkpoints in terms of training step. We combine both these insights to induce diversity in our checkpoints.

## 2.3 LAWA: LATEST WEIGHT AVERAGING

We explain the Latest Weight Averaging (LAWA) algorithm below along with a python-style pseudo code (Algorithm 1). As shown in Figure 3, LAWA maintains a first in first out (FIFO) queue of periodically sampled checkpoints with a large number of intervening steps ($\nu$) in between two succesive samples. We adapt LAWA for our setting with minor modifications. Specifically, we introduce k_stepsize ($\nu$), and decoupled interval and k to effectively sample distant checkpoints in the training run. The original LAWA algorithm (Kaddour, 2022) assumes interval = k.

> LAWA runs a moving window at a predetermined interval to collect k latest checkpoints on sequence of saved checkpoints $\theta_t$. The LAWA derived checkpoints are computed as $\theta_t^{\text{LAWA}} := \frac{1}{k} \sum_{s=t-k}^{t} \theta_s$ where $\theta_t$ the original checkpoints are sampled several training steps apart in the training process.

## 3 EXPERIMENTAL SETUP

**NanoGPT-2 Experiments.** We conduct all our experiments utilizing autoregressive decoder-style Large Language Models (LLMs), specifically nanoGPT-2 and Pythia LLMs. We utilize three distinct sizes of nanoGPT-2 models: small (125M), medium (355M), and large (770M). We train nanoGPT-2 models from scratch using the OpenWebText dataset, which includes 9 billion training tokens and 4.4 million validation tokens. Throughout the experiments, we maintain a consistent sequence length of 1024 and a fixed batch size of 131K tokens per batch, the latter being the maximum batch size accommodated by our GPUs. The configurations for the model and pre-training were adapted from Sophia's (Liu et al., 2023) AdamW baseline, with adjustments made to the learning rate and batch size to align with our specific needs. Notably, we trained all the models with learning rates

---

**Algorithm 1** LAWA: Pytorch-style pseudocode

```
def LAWA(ckpts, interval, k, k_stepsize)
    :
    # ckpts: list of checkpoints
    # k: number of checkpoints to average
    # k_stepsize: distance in training
        steps between checkpoints
    averaged_ckpts = []
    for i in range(0, len(ckpts),
        interval):
        start_idx = i - k
        end_idx = i - 1
        # select last k checkpoints
        # with k_stepsize between each
            checkpoint.
        selected_ckpts = ckpts[end_idx-k
            +1:end_idx+1:k_stepsize]
        avg = average(selected_ckpts)
        averaged_ckpts.append(avg)
    return averaged_ckpts
```

---

that were ten times higher and batch sizes that were twice as large compared to the configurations in Liu et al. (2023), where the learning rate was tuned through a grid search. We compare LAWA with the original pre-training recipe, EMA (Szegedy et al., 2015), and SWA (Izmailov et al., 2018), which we adapt for LLMs. For EMA, we set the decay to 0.9 as per Kaddour (2022) and update the EMA model at every step, which is a standard practice. For SWA, we adhere to the original pre-training procedure until 75% completion, after which SWA training is initiated with a new SWA scheduler (cosine annealing). We compute the SWA uniform average every 10 steps.

**Pythia Experiments.** The **Pythia LLMs** are publicly available in the Pythia suite (Biderman et al., 2023). We report results on Pythia-1B, Pythia-2.8B, Pythia-6.9B, and Pythia-12B; Table.1 summarizes the details of these models. For our experiments, we use intermediate model checkpoints; saved after every 1000 update steps. The models are trained by Biderman et al. (2023) on the PILE dataset (Gao et al., 2020), a publicly available, curated collection of English text corpus of size 800GB. The original PILE dataset is curated using 5 different genres of data namely, academia, internet, prose,

| Model Size | Layers | Hidden Size | Heads | Learning Rate | Equivalent Models |
|---|---|---|---|---|---|
| 125M | 12 | 12 | 768 | $6.0 \times 10^{-3}$ | nanoGPT-2 (small) |
| 335M | 24 | 1024 | 16 | $3.0 \times 10^{-3}$ | nanoGPT-2 (medium) |
| 770 M | 36 | 1280 | 20 | $2.0 \times 10^{-3}$ | nanoGPT-2 (large) |
| 1.0 B | 16 | 2048 | 8 | $3.0 \times 10^{-4}$ | — |
| 2.8 B | 32 | 2560 | 32 | $1.6 \times 10^{-4}$ | GPT-Neo 2.7B, OPT-2.7B |
| 6.9 B | 32 | 4096 | 32 | $1.2 \times 10^{-4}$ | OPT-6.7B |
| 12 B | 36 | 5120 | 40 | $1.2 \times 10^{-4}$ | — |

Table 1: Overview of models and their architecture from the nanoGPT-2 suite and Pythia suite (Biderman et al., 2023) used in our experiments. The model nomenclature for Pythia LLMs is pythia-xx with model size. Models marked as "equivalent" have the same architecture and number of non-embedding parameters.

dialogue and miscellaneous. The PILE dataset contains 300 billion tokens prior to deduplication, and this number reduces to 207 billion tokens after the deduplication process. Our experiments use Pythia models trained with PILE-deduped dataset, as such models tend to memorize less (Lee et al., 2021). The batch size for all the Pythia models was set at 2.09 million tokens and the learning rate was scaled following Zhang et al. (2022).

**Evaluation**   We evaluate the language modelling performance of nanoGPT-2 models pre-trained for 70K steps using log perplexity (perplexity and loss used interchangeably) on the held-out/val set. For nanoGPT-2 we use the moving window interval = 1K, k = 5 and we sample checkpoints interval = 200,1K apart for LAWA. Next we analyze the original training trajectories of Pythia LLMs and demonstrate the improvements achieved in test generalization using LAWA. We also present zero-shot evaluation results on Lambada OpenAI (Paperno et al., 2016), SciQ (Welbl et al., 2017), AI2 Reasoning Challenge-easy (ARC-e) (Clark et al., 2018), and Wikitext (Merity et al., 2016). We evaluate 4 Pythia LLMs using the intermediate model checkpoints on a subset of the test and validation dataset following the methodology prescribed by Xia et al. (2022). For the purpose of evaluation we use the open source library `lm-evaluation harness`[2]. We select representative subsets from the diverse genres encompassed by the full PILE validation and test dataset. This subset comprises PILE-philosophy papers, PILE-bookcorpus2, and PILE-YouTube subtitles datasets.

We conduct zero-shot evaluation on the Pythia LLMs. For zero-shot evaluation we provide a natural language description of the downstream task, along with a textual example. The models then generate responses that are either open-ended or discriminatively select a proposed answer. This evaluation setup serves as a robust academic benchmark, as it assesses Pythia models of various scales on reasonably large PILE subsets and downstream datasets, both in terms of test performance and zero-shot. For both the test generalization and zero-shot experiments, we evaluate model checkpoints starting 21K steps to 141K steps (recall that subsequent checkpoints are 1K steps apart). Moreover, we choose to slide the averaging window at 3K steps (i.e. interval = 3K) and average last k intermediate checkpoints as discussed in LAWA algorithm (Algorithm 1). Our selection of LAWA parameters such as k = 5 and start step = 21K are based on the experiments discussed in Section A.1.

## 4 RESULTS

### 4.1 EXPLORING LLM PRE-TRAINING WITH NANOGPT-2 AT SMALL SCALE

**LLMs trained with higher LRs observe higher gains with LAWA.**   We ran controlled experiments to better understand the correlation between LR and gains due to checkpoint averaging. Initially we train nanoGPT-2 small with two different LRs ($6 \times 10^{-3}, 6 \times 10^{-4}$), keeping batch size and all relevant hyperparameters the same. $6 \times 10^{-4}$ is the assumed optimal LR computed using a grid search reported in Liu et al. (2023). Subsequently, we pre-trained the same model using an LR of $6 \times 10^{-3}$, which is tenfold higher in magnitude than the former. As shown in Figure 2(a), models

---

[2]https://github.com/EleutherAI/lm-evaluation-harness

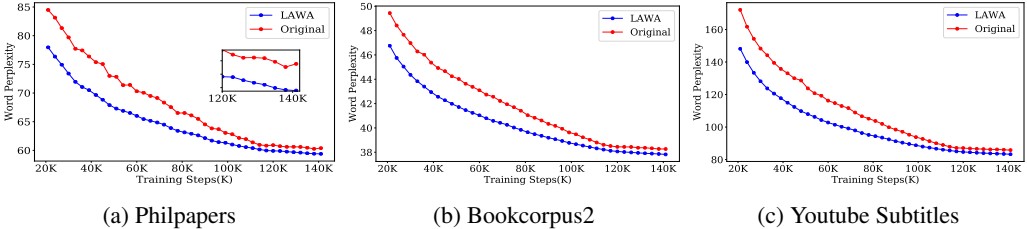

| (a) Philpapers | (b) Bookcorpus2 | (c) Youtube Subtitles |

Figure 4: **LAWA speeds up convergence for Pythia-1B on subset of tasks from the original pretraining dataset i.e. PILE**. We present the original and the LAWA training trajectories for 3 different tasks from PILE namely philpapers, bookcorpus2 and youtube subtitles.

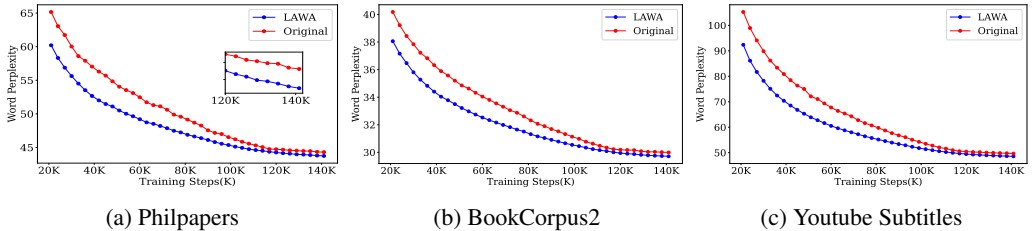

| (a) Philpapers | (b) BookCorpus2 | (c) Youtube Subtitles |

Figure 5: **LAWA speeds up convergence for Pythia-2.8B on subset of tasks from the original pretraining dataset i.e. PILE.**. We present the original and the LAWA training trajectories for 3 different tasks from PILE namely philpapers, bookcorpus2 and youtube subtitles.

trained with higher LR observe higher gains compared to its counterpart trained with lower LR due to post hoc checkpoint averaging in LAWA. From Figure 2(b) we observe that the model trained with a higher LR converges faster but compromises on generalization, a phenomenon also observed by Kaur et al. (2022). The gap in generalization is effectively mitigated by checkpoint averaging through LAWA, as shown in Figure 2(c). Interestingly, we note that the training trajectory of LAWA approximates that of a model trained with a lower LR. This is an important insight: checkpoint averaging acts as a surrogate for LR decay, thereby enabling the model to be trained with a higher LR. In practical LLM pre-training scenarios, where conducting a grid search is challenging due to the model's size, adopting our proposed training recipe could be advantageous. One might select a higher LR (that doesn't cause divergence) and train an LLM faster without compromising much generalization compared to conventional pre-training strategy.

**LAWA improves test generalization in fewer training steps compared to original pre-training and relevant baselines.** LAWA clearly outperforms the original pre-training run starting very early on during training, as shown in Figure 1. Since we employed a reasonably high LRs for all the nanoGPT-2 LLMs, we observe higher gains in the early-mid stages of pre-training, and the gains start diminishing towards the final stages due to the LR scheduler that continuously decays the weight throughout the training cycle. Additionally LAWA also outperforms important baselines such as EMA and SWA. LAWA clearly has an edge over EMA throughout all training phases. Our experiments also reveal that applying SWA during the early stages of training leads to divergence (Figure 11). Consequently, LAWA outshines SWA in both performance and ease of implementation.

**The gains due to LAWA amplifies with far checkpoint averaging.** As shown in Figure 1, LAWA with higher k_stepsize ($\nu$) performs better particularly for larger models. Intuitively, we believe that the diversity between nearby checkpoints might be very low given that larger

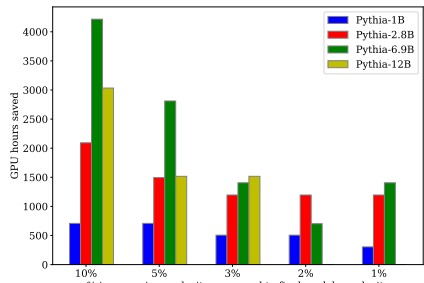

Figure 6: LAWA saves significant amount of GPU hours compared to original training. We compare the savings in GPU hours as a function of increase in final perplexity, i.e. perplexity achieved at 141K training step by the original checkpoint. This plot is created using a held out set from the training subset PILE-philosophy papers.

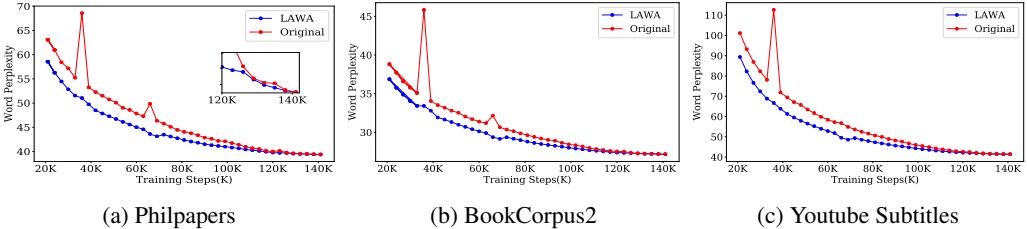

(a) Philpapers  (b) BookCorpus2  (c) Youtube Subtitles

Figure 7: **LAWA speeds up convergence for Pythia-6.9B on subset of tasks from the original pretraining dataset i.e. PILE**. We present the original and the LAWA training trajectories for 3 different tasks from PILE namely philpapers, bookcorpus2 and youtube subtitles.

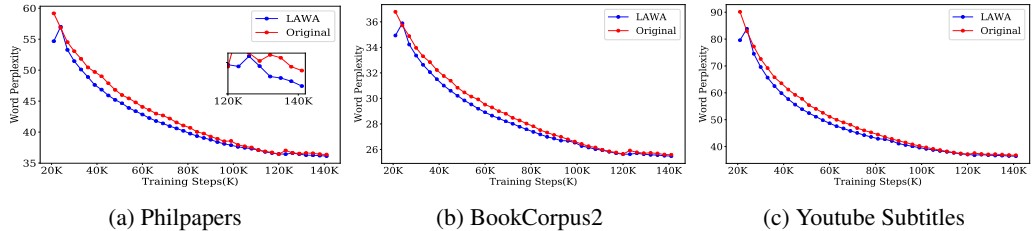

(a) Philpapers  (b) BookCorpus2  (c) Youtube Subtitles

Figure 8: **LAWA speeds up convergence for Pythia-12B on subset of tasks from the original pretraining dataset i.e. PILE**. We present the original and the LAWA training trajectories for 3 different tasks from PILE namely philpapers, bookcorpus2 and youtube subtitles.

models learn faster (Li et al., 2020). Hence, one needs to sample more distant checkpoints for larger models. This observation is also consistent with billion-parameter Pythia LLMs, as shown in Figure 10.

## 4.2    SCALING TO BILLION PARAMETER PYTHIA LLMS

Figures 4 and 5 illustrate that the checkpoints derived using LAWA demonstrate better test generalization than the checkpoints saved during original training for the Pythia-1B and Pythia-2.8B models i.e. (moderate size LLMs). In Figures 7 and 8, we observe significant improvements in test generalization during early-mid training regime and minor improvements towards the end for both Pythia-6.9B and Pythia-12B models. All the LAWA LLMs achieve lower perplexity with lesser training steps compared to the original training trajectory, thereby saving significant amount of GPU hours (Figure 6), subsequent training costs and ingested training data. The savings are computed based on Table 3 in the appendix. Additionally, LAWA proves beneficial in situations where training similar models from scratch is necessary but can only be conducted over a limited number of training steps due to strict compute budgets.

To analyze the phenomenon wherein moderate size LLMs exibit higher gains compared to their larger counterparts in test performance across both early-mid and final training trajectories, we delve into the Pythia suite's training methodologies. The authors of Pythia suite (Biderman et al., 2023) report that they have used an exceptionally large batch size (2M tokens) and learning rate for Pythia-1B and Pythia-2.8B models to expedite the convergence. For the larger Pythia models, such as Pythia-6.9B and Pythia-12B, learning rates are reduced to $1.2 \times 10^{-4}$ while maintaining the batch sizes to 2M tokens, in line with prior work. Overall we observe thematically similar trends with the nanoGPT-2 experiments.

## 4.3    MITIGATION OF LOSS SPIKES DURING EVALUATION

Recent work (Kaplan et al., 2020; Zhang et al., 2022; Touvron et al., 2023a; Dey et al., 2023; Wortsman et al., 2023) report loss spikes– brief degradations in the performance along a training trajectory when scaling up the model size, batch size, and learning rate. In our evaluations, we observe two perplexity spikes (Figure 7). Interestingly, we find that LAWA mitigates the spikes during evaluation quite effectively. This can be intuitively explained by the **smoothing effect** – average out the outliers to fit the trend – resulting from averaging checkpoint weights over a range

| Models | Steps | Lambada openai | | SciQ | | WikiText(↓) | | ARC-easy | |
|---|---|---|---|---|---|---|---|---|---|
| | | LAWA | Original | LAWA | Original | LAWA | Original | LAWA | Original |
| Pythia-1B | 48 K | 50.32 | 46.85 | 84.6 | 84.3 | 18.34 | 19.33 | 54.50 | 54.25 |
| | 60 K | 50.77 | 47.00 | 84.6 | 84.4 | 17.91 | 18.82 | 55.18 | 54.50 |
| | 105 K | 57.84 | 56.39 | 86.1 | 86.3 | 16.83 | 17.12 | 56.77 | 56.27 |
| | 141 K | **58.99** | 58.68 | 86.7 | 87.6 | **16.50** | 16.71 | **58.33** | 58.16 |
| Pythia-2.8B | 48 K | 63.5 | 61.9 | 86.5 | 85.6 | 14.60 | 15.37 | 61.4 | 60.6 |
| | 60 K | 64.3 | 63.8 | 86.8 | 86.3 | 14.17 | 14.87 | 62.7 | 62.1 |
| | 105 K | 64.77 | 63.14 | 87.7 | 87.4 | 12.91 | 13.08 | 63.67 | 63.04 |
| | 141 K | **65.47** | 65.26 | **88.8** | 88.6 | **12.59** | 12.70 | **64.68** | 64.56 |
| Pythia-6.9B | 48 K | 65.8 | 62.3 | 88.7 | 88.0 | 13.55 | 14.25 | 63.7 | 62.9 |
| | 60 K | 67.1 | 64.6 | 88.6 | 89.0 | 13.04 | 13.61 | 64.1 | 62.9 |
| | 105 K | **68.05** | 67.78 | **91.1** | 91.2 | 11.92 | 12.07 | **67.88** | 67.08 |
| | 141 K | **69.08** | 68.85 | **92.0** | 91.7 | **11.61** | 11.70 | **68.13** | 67.80 |
| Pythia-12B | 48 K | 66.8 | 65.4 | 89.7 | 88.8 | 13.09 | 13.35 | 66.4 | 65.1 |
| | 60 K | 67.8 | 66.2 | 90.3 | 90.5 | 12.54 | 12.76 | 67.3 | 60.1 |
| | 105 K | **71.06** | 70.65 | 91.6 | 91.9 | 11.17 | 11.33 | 69.78 | 69.31 |
| | 141 K | **71.56** | 71.00 | **92.8** | 92.3 | **10.84** | 10.91 | 70.58 | 70.74 |

Table 2: LAWA improves zero shot performance on academic question answering and knowledge assessment downstream tasks starting very early on in the training. The checkpoints derived using LAWA requires less steps to reach higher zero-shot performance than the checkpoints derived using original training. We indicate the scores in bold font when the performance of LAWA surpasses the final score (at 141K steps) obtained using the original training or achieves comparable performance significantly earlier, specifically at 105K steps.

of steps that are far apart. Note that since we have sampled the checkpoints at an interval of 3K for LAWA, we may have inadvertently overlooked some checkpoints exhibiting loss spikes in models other than Pythia-6.9B.

## 4.4 IMPROVED ZERO-SHOT PERFORMANCE

LAWA improves the zero-shot performance in several ways. First, we observe that zero-shot performance of early-mid checkpoints (48K, 60K) achieves higher performance almost consistently, regardless of the scale as shown in Table 2. For instance, the LAWA Pythia-1B checkpoint evaluated at 24K steps on the Lambada OpenAI task achieves higher accuracy than the original checkpoint evaluated at the 48K step.

We note that the checkpoints derived using LAWA also exhibit improvements in the later stages of training (105K,141K) on the majority of tasks, highlighted by the bolded numbers in Table 2. Moreover, we consistently witness gains until 105K steps across all models, which constitutes approximately 75% of the total training steps. Therefore our recipe proves to be beneficial in a compute optimal LLM training scenario where early stopping is employed at 75% of total training. Additionally, we find that our LAWA derived checkpoints of Pythia-6.9B reach the final accuracy/perplexity on multiple downstream tasks considerably earlier, specifically at the 105K step mark. Intuitively, we know that higher zero-shot performance on various different downstream tasks is associated with low perplexity in language modelling during training, a correlation that is mathematically substantiated (Saunshi et al., 2020). Therefore, all the observations we made in Section 4.1 naturally apply to zero-shot performance as well.

## 4.5 DIFFUSION MODELS

We also experiment with image diffusion models to gauge the effectiveness of LAWA on generative models beyond language. The underlying model is a 422M parameter UNet (Ronneberger et al., 2015; Ho et al., 2020) trained with $\epsilon$-prediction objective and standard cosine schedule (Ho et al., 2020) on ImageNet 128x128 dataset. The model was trained with the Adam (Kingma & Ba, 2014)

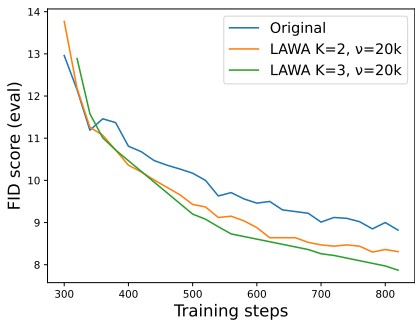

Figure 9: LAWA speeds up the convergence for Image diffusion model, measured in terms of FID on the evaluation set for ImageNet-128x128.

optimizer using a learning rate of 1e-4. Figure 9 shows the FID on the evaluation set for the baseline checkpoints and LAWA averaging over the baseline checkpoints. Note that the baseline checkpoints themselves are obtained using the Exponential Moving Average (EMA) with decay rate of 0.9999 over the training trajectory, following the standard practice in training diffusion models. It is noteworthy that LAWA checkpoint averaging improves the FID over the already EMA'ed checkpoints. We defer a more thorough empirical investigation of LAWA for the family of diffusion models to future work.

## 5 RELATED WORK

**Weight Averaging (WA)** has been studied and employed since the 1960s, predominantly in simple linear (Lakshminarayanan & Szepesvari, 2018) and convex settings (Polyak & Juditsky, 1992; Neu & Rosasco, 2018). Recent WA approaches in deep learning can be broadly classified into two categories; First, approaches that simultaneously trains multiple models with different initialization and hyper-parameters (Wortsman et al., 2022; Rame et al., 2022; Jolicoeur-Martineau et al., 2023; Rame et al.) to later average them for better generalization. Second, approaches that focus on improving generalization of the final model or models close to convergence (Tarvainen & Valpola, 2017; Izmailov et al., 2018; Athiwaratkun et al., 2018b; Yang et al., 2019; Cha et al., 2021). Stochastic WA (SWA) (Izmailov et al., 2018) employs a similar technique of averaging checkpoints along training trajectories but only works in the later stages of training (i.e. post 75% of the training run) with a new LR scheduler. This unsual halting and restarting the training with SWA with a new schedular limits its integration. We also show that SWA when applied early on during training diverges (Section A.2). Our recipe focuses on getting early gains through early averaging and can be generally applied to a wide range of training regimes.

## 6 CONCLUSION AND FUTURE WORK

In this paper we investigated a LLM pre-training setting where the LR is significantly higher than what is conventionally used. This scenario is particularly practical as LLMs are often trained using numerous GPUs in parallel, necessitating higher batch sizes for optimal GPU utilization. Here we introduce early weight averaging throughout the training trajectory utilizing LAWA. Our findings indicate that this strategy enables LLMs to generalize more effectively in fewer steps compared to the original pre-training scheme, and key baselines as demonstrated using nanoGPT-2 models. Subsequently, we applied LAWA to Pythia LLMs of varying scales—1B, 2.8B, 6.9B, and 12B parameters—to assess its effects. Remarkably, we observed consistent gains in both test and zero-shot performance across different scales. Similar performance enhancements were also noted when LAWA was applied to generative image diffusion models. Looking forward, several fascinating avenues for future research are evident. These encompass the application of proposed pre-training recipe in a fine-tuning scenario with a very restricted training budget and continual training of the intermediate averaged checkpoints.

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

# Supplementary Materials: Appendix

## CONTENTS

## A   SUPPLEMENTARY EXPERIMENTS AND RESULTS

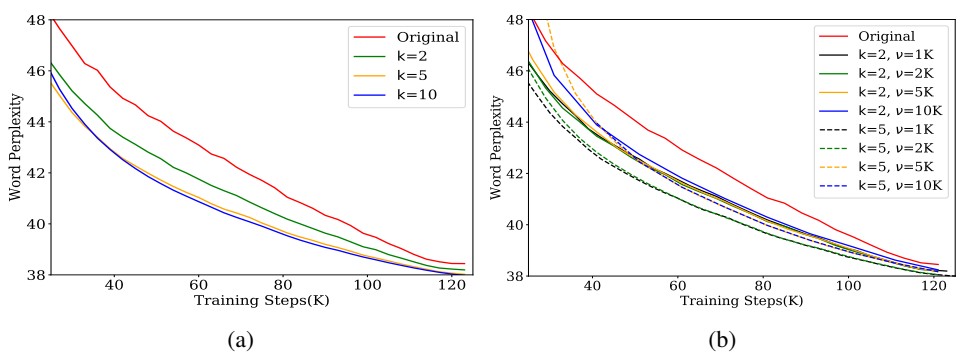

(a)                                                (b)

Figure 10: Ablations studying test performance as a function of (a) number of checkpoints to be averaged $k = \{2, 5, 10\}$ at $\nu = 1K$, (b) distance of checkpoints to be averaged $\nu = \{1K, 2K, 5K, 10K\}$ at $k = \{2, 5\}$.

### A.1   ABLATIONS

We perform ablations to better understand the interplay between the performance of number of checkpoints to be averaged $\theta_t^{\text{LAWA}}$ with varying $k$ and distance between averaged checkpoints $\nu$ in training. We study the Pythia-1B model with held-out subset of PILE-bookcorpus2. Additionally, we also provide training trajectory of Pythia-1B model on a subset of PILE datasets much earlier than 21K steps (refer Section A).

**Test Performance with varying $k$ and fixed $\nu = 1$K.**   We investigate the impact of varying $k$ on the model's test performance, while keeping $\nu$ constant at 1K. Our aim is to determine the optimal number of checkpoints to include in the average when selecting the latest $k$ checkpoints following the LAWA approach. As outlined in Section 3, the Pythia checkpoints are saved at a frequency of 1K, so we have maintained $\nu$ at 1K for this analysis. From Figure 10(a), it is apparent that a smaller $k$ could be detrimental, but performance remains fairly stable for reasonably large $k$ values. Consequently, for our LLM experiments, we opted for $k = 5$ since $k = 10$ tends to occupy a substantial amount of disk space, especially for larger models such as Pythia-12B.

**Test Performance with varying $\nu$ for $k = \{2, 5\}$.**   Memory requirements remain a key bottleneck in saving model checkpoints throughout the training trajectory particularly for extremely large billion parameter models. Therefore, it is very interesting to know how far away checkpoints in a training trajectory can be averaged? We investigate this question with far checkpoint averaging at $\nu = \{1K, 2K, 5K, 10K\}$ training steps apart for $k = \{2, 5\}$. As shown in Figure 10(b), we observe for both $k = 2$ and $k = 5$, we find that averaging more recent checkpoints (keeping $\nu$ small) works better than averaging stale weights (higher $\nu$). For instance LAWA using $k = 5$ and $\nu = \{1K, 2K\}$ consistently performed better than against LAWA with other parameters. Overall, we empirically find that a moderate number of checkpoints ($k = 5$) saved in smaller frequencies ($\nu = 1K$) works best.

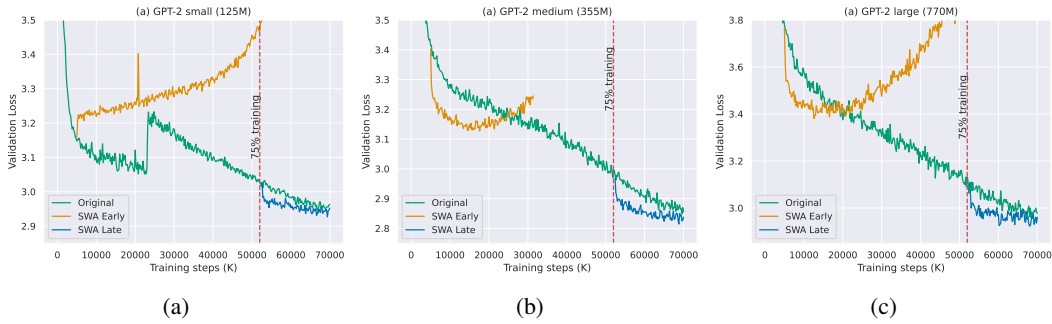

Figure 11: Across all model sizes (125M, 355M, 770M), we observe that the early version of stochastic weight averaging diverges, in contrast to the originally proposed late version.

## A.2 EARLY-SWA EXPERIMENTS

Stochastic Weight Averaging Izmailov et al. (2018) has previously shown gains for smaller models, particularly in late stages of training (typically $> 75\%$). We experimented with initializing SWA in early stages of training. As shown in Figure 11, we observe that it diverges quickly - showing that SWA does not provide any gains earlier in training.

---

**Algorithm 2** Pytorch-style pseudocode of EMA/SWA

```python
def EMA_SWA(ckpt, alpha, step_size, init_point):
    """
    ckpt: list of data points (could be checkpoints or any data series)
    alpha: smoothing factor, 0 < alpha <= 1. If alpha < 0, enables SWA.
    step_size: How often to calculate average. Typically set to 1 for EMA.
    init_point: Step after which to start averaging. Typically 0 for EMA.
    """
    # Initialize the series with the first data point
    series = [ckpt[0]]
    n_models = 1
    for i in range(1, len(ckpt)):
        # Calculate EMA/SWA
        if i%step_size==0 and i>init_point:
            if alpha < 0:
                factor = n_models/(n_models + 1)
                value = (1 - factor) * series[i-1] + factor * ckpt[i]
                n_models += 1
            else:
                value = (1 - alpha) * series[i-1] + alpha * ckpt[i]
            series.append(value)

    return series
```

---

## A.3 PHASE TRANSITION AND LINEAR MODE CONNECTIVITY

Averaging very initially i.e. before 8K steps during training may not always yield beneficial results (Figure 12). However, the technique does start showing efficacy fairly early in the training process. We highlight this phenomenon by presenting experimental results with the Pythia-1B model using a held-out set of PILE-bookcorpus2 and PILE-enron emails. We observe that LAWA trajectory undergoes a phase transition at the 8K training step. Beyond this transition, significant improvements in test performance can be seen. Such a phase transition may not occur for all Pythia LLMs. Following this phenomenon we presented our results starting 21K steps in Figures 4-8. We further examine this phenomenon through the lens of linear mode connectivity.

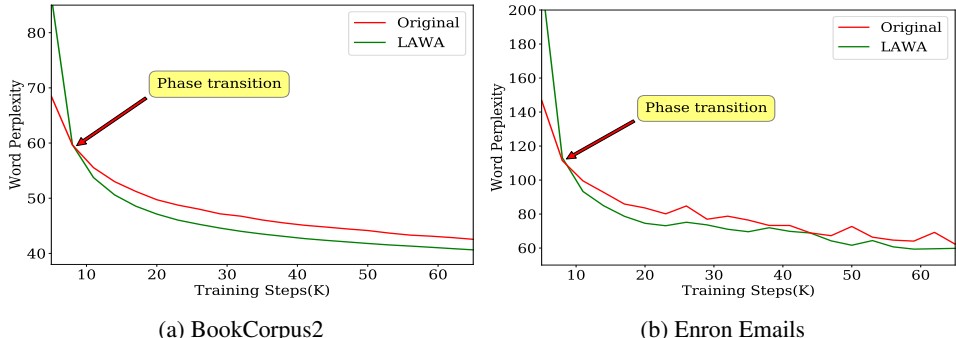

(a) BookCorpus2          (b) Enron Emails

Figure 12: **Early weight averaging doesn't work at the very beginning of the training but works reasonably early during the training process.** Here we compare original and LAWA early training trajectories for Pythia-1B model on 2 different tasks namely bookcorpus2 and enron emails using held out set.

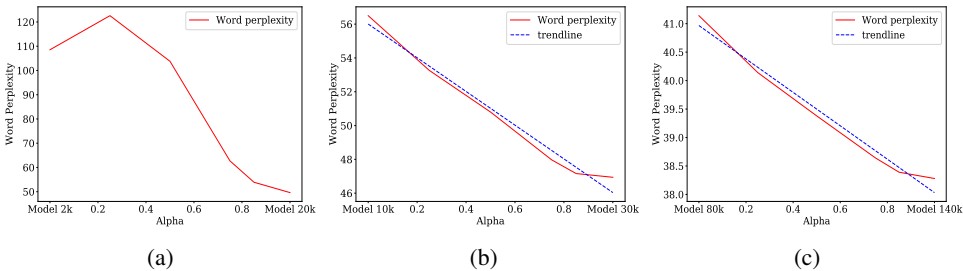

(a)          (b)          (c)

Figure 13: **The model checkpoints attains linear mode connectivity quite early but not at the very beginning of the training process.** We plot word perplexity as a function of the model derived from the convex combination of 2 different checkpoints i.e. $\boldsymbol{\theta}_{\text{LMC}}$ at $\alpha = \{0, 0.2, 0.4, 0.6, 0.8, 1\}$. In (a) we see an error barrier that means model checkpoint at 2K and 20K are not linear mode connected, whereas both (b) and (c) shows the checkpoints under consideration are linear model connected.

To better comprehend the linear model connectivity of checkpoints, we perform a convex combination of model checkpoints at different training stages. For instance, a model checkpoint at 2K and 20K can be combined in this manner: $\alpha \times \boldsymbol{\theta}_{2k} + (1 - \alpha) \times \boldsymbol{\theta}_{20k}$. In Fig. 10, we plot word perplexity as a function of $\alpha$ using PILE-bookcorpus2. Here we observe that initially the model checkpoints are not linear mode connected. However, based on the evaluated checkpoints shown in Figure 13, we posit that the model checkpoint attains linear mode connectivity (LMC) quite early and maintains this property until the end of training.

# B   AMOUNT OF COMPUTE

We compute the savings in GPU hours based on the Table. 6 of Pythia suite Biderman et al. (2023) as shown below.

| Model Size | GPU Count | Total GPU hours required |
|:---:|:---:|:---:|
| 1.0 B | 64 | 4,830 |
| 2.8 B | 64 | 14,240 |
| 6.9 B | 128 | 33,500 |
| 12 B | 256 | 72,300 |
| Total | | 136,070 |

Table 3: Table from Biderman et al. (2023). Model sizes in the Pythia suite, number of GPUs used during training, and the total number of GPU hours, calculated as (iteration time (s) $\times$ number of iterations $\times$ number of GPUs $\div$ 3600 s/hour). All GPUs are A100s with 40GB of memory.

