# OpenReview forum: "Early Weight Averaging Meets High Learning Rates for LLM Pre-training"
_ICLR.cc/2024/Conference — ICLR 2024 Conference Withdrawn Submission_

### Official Review · Reviewer_EwGY · 2023-11-03

**Soundness:** 3 good
**Presentation:** 3 good
**Contribution:** 1 poor
**Rating:** 3
**Confidence:** 4

**Summary:**

This paper proposes to improve the convergence speed of training large language models and diffusion models by increasing the learning rate, and averaging checkpoints early in training to stabilize the training and improve overall result. The authors compare with using EMA checkpoints and standard training and report that, at least at smaller scales, their approach results in faster convergence and better models on some of the studied benchmarks.

**Strengths:**

- Thorough ablations and study of independent variables.
- Some of the experiments at smaller and medium regimes look promising.

**Weaknesses:**

- The paper’s contribution is limited, and can be viewed as specialization of LAWA (Kaddour et al, 2022) to decoder only transformers by picking specific hyperparameters. Note that Kaddour et al already performs experiments on Roberta.
- The proposed method does not scale (Figures 6-8), and demonstrates much better performance for medium sized models than large ones. This makes it unlikely that this technique will help in the large model regime that the community really cares about.
- The intuitive explanation for the optimization viewpoint (Sec 2.2) doesn’t make sense, and ignores the fact that the entire point of Adam and second order optimizers is to precondition the updates. Furthermore, in their heuristic pictures, simply using a momentum optimizer should give similar acceleration as LAWA.

**Questions:**

- The paper frequently mentions a generalization gap. In LLM training with less than 1 epoch, it is rare to get an actual generalization gap. This is especially the case when the model is small (770M is the largest model trained in Fig 1 for example) and datasets are large (PILE). I am very confused that one of the claimed observations is that “the generalization gap is decreased.” Can you please explain?
- One justification of using LAWA is that you can train with higher lr. A priori, this isn’t a desirable thing e.g., is it clear in Figure 2 that you can’t recover the LAWA results by just lowering lr below 6e-4?
- Please clarify whether the baseline experiments are already trained with lr decay.

---

### Official Review · Reviewer_Y53G · 2023-11-05

**Soundness:** 3 good
**Presentation:** 3 good
**Contribution:** 1 poor
**Rating:** 3
**Confidence:** 3

**Summary:**

This paper proposes to adapt LAWA (LAtest Weight Averaging) to a large language model pre-training. The paper first describes the benefits of LAWA-based model averaging in terms of stable training, even in large LR cases, and ensemble effects. Then, the paper intensively examines the effectiveness of LAWA with nanoGPT and Pythia-based LLMs and confirms the generalization capability of the proposed methods and the convergence properties. The paper also applies this method to image-based diffusion models successfully.

**Strengths:**

- Investigation of LLMs is a very hot topic in AI. This paper investigates the fast convergence and generalizability of LLMs by using LAWA.
- It is highly reproducible (open source, public data, and clear descriptions of hyper-parameter configurations)
- The paper has a lot of analyses (convergence in terms of LRs, model sizes, different LLMs, zero-shot capabilities, and application to image-based diffusion models), and it is worth sharing them with the community.
- Section 2.2 has very clear and intuitive explanations.

**Weaknesses:**

- The novelty is weak. This paper's technical novelty from LAWA (LAtest Weight Averaging) (Kaddour, 2022) is very minor.
- The paper also lacks the theory to justify the discussions.
- Although we observe the fast convergence, it converges to a similar performance to the conventional methods in similar iterations (e.g., the proposed method converges to similar values in the similar iterations in Figures 4-8. The performance improvement in Tables is marginal). Thus, I could not find the benefits of the proposed method.

**Questions:**

- Does Figure 2-(b) and Figure 2-(c) come from Figure 2-(a)? If so, it is redundant, and you can discuss it only with Figure 2-(a).
- Section 3 "We evaluate the language modelling performance of nanoGPT-2 models pre-trained for 70K steps": 70 K is too small to discuss the convergence. Also, the later experiments used 140K, and the experimental conditions were inconsistent. Can you clarify why you use 70K?
- Algorithm 1: Can you add more explanations of how to use multiple averaged checkpoints in the list?
- Does Figure 6 compare with the conventional training methods? I could not understand how to check the computational efficiency of the proposed method compared with them from Figure 6.

---

### Official Review · Reviewer_eRWc · 2023-11-06

**Soundness:** 3 good
**Presentation:** 2 fair
**Contribution:** 2 fair
**Rating:** 3
**Confidence:** 3

**Summary:**

This paper empirically demonstrates that models trained with high learning rates show greater improvements over original training and this gain is further amplified when sampling distant checkpoints. Integrating these two insights with a previous method LAWA (Latest Weight Averaging), this paper accelerates the training process on models with different scales, ranging from 125 million to 12 billion parameters. Experiments show that the method proposed improves test and zero-shot generalization while accelerating the convergence.

**Strengths:**

1.	This paper is well-written and provides source code for re-implement.

2.	The method is simple and effective. Experiments show that LAWA with a higher learning rate speeds up convergence and improves the zero-shot generalization performance.

**Weaknesses:**

1.	This paper does not compare the zero-shot performance of SWA, and EMA. Will higher learning rate and distant checkpoint sampling still work on these weight averaging methods?

2.	The improvements in zero-shot performance seem a little marginal.

3.	This paper seems to be a combination of LAWA and a high learning rate with an average weight of distant checkpoints, I am wondering if this achieves ICLR’s novelty bar.

4.	This paper only tests the zero-shot performance. What about other scenarios? If models pre-trained with the approach proposed, can not only perform better in a zero-shot setting but also better after instruction-tuning, fine-tuning, etc., I would consider raising my score.

**Questions:**

1.	This paper has been accepted by the NeurIPS 2023 workshop WANT.  Not sure whether it follows the submission policy of ICLR.

2.	Typos
a.	Contribution 1, we provide a intuitive …  -> we provide an intuitive …
b.	Conclusion, a LLM pretraining -> an LLM pretraining

---

### Official Review · Reviewer_7tZF · 2023-11-08

**Soundness:** 1 poor
**Presentation:** 2 fair
**Contribution:** 1 poor
**Rating:** 3
**Confidence:** 4

**Summary:**

In this paper, authors tackle a problem of improving the autoregressive Large Language Models (LLMs) pre-training characteristics (log perplexity), while increasing the efficiency (cost / training steps / GPU hours) of pre-training at the same time. To achieve this goal they test different methods for checkpoint averaging (LAWA - Latest Weight Averaging, EMA - exponential moving average, SWA - stochastic moving average) starting early on during training. They evaluate, using both perplexity and downstream task evaluation, these methods on Transformer decoder-only models of various scales ranging from 125M to 12B parameters.

Contributions:
- They show that if you use checkpoint averaging methods like LAWA you can train LLMs with high LR without compromising generalisation (Figure 2)
- They demonstrate efficiency and performance (intermediate) gains (Figure 1, Table 2), for language models (nanoGPT-2, Pythia) and diffusion model for image generation (Figure 9), when trained with checkpoint averaging.

**Strengths:**

Originality:
- The subfield of accelerating the training of LLMs is very broad and vital, but in this work authors test narrow and not yet tested (?) approach, i.e., checkpoint averaging to improve the efficiency. Furthermore authors test this idea on diffusion model for image generation.

Quality & Clarity:
- I believe that the paper is generally well written and nicely structured. It’s nice that authors performed experiments on a wide range of LLMs ranging from 125M to 12B parameters.
- I appreciate that authors present a wide evaluation, not only using language modelling perplexity, but also some range of downstream tasks which do not always correlate with perplexity.

Significance:
- Training LLMs is very crucial these days so any work on making this more efficient is significant in this sense.
- I think that this work is significant in that sense that it studies the application of checkpoint averaging, which is a widely known technique and some people probably were thinking of testing it themselves, to LLM pre-training

**Weaknesses:**

Artificial setup:
- “To accelerate the training process, it is a popular practice in LLM pre-training to utilize exceptionally large batch sizes, thereby ensuring maximal GPU utilization and consequently necessitating the use of relatively high learning rates.” - I agree that in the current LLM pre-training we are using large batch sizes to better utilise hardware, but I think that the paper lacks evidence of how it leads to “relatively high learning rates”. We are using large batch sizes, but at the same time we are increasing the amount of training data, therefore we have enough training steps so that we do not need to significantly scale up learning rate. For the LLaMA2 paper authors used 3e-4 and 1.5e-4 learning rates which are pretty standard. Pythia checkpoints which you are using were trained with LRs lesser than 1.2e-4. Similar LRs were used by the OPT LLMs.
- “Since we employed a reasonably high LRs for all the nanoGPT-2 LLMs, we observe higher gains in the early-mid stages of pre-training, and the gains start diminishing towards the final stages due to the LR scheduler that continuously decays the weight throughout the training cycle. - I don’t think this is fair to compare validation loss in the early-mid stages of pre-training while the LR hasn’t been fully decayed. Instead of comparing the validation loss after a given amount of steps, where the LR is not properly annealed yet, more fair comparison would be to tune both baseline and the proposed method for some pre-defined compute budget and then report the improvements.
- “LLMs trained with higher LRs observe higher gains with LAWA. […] 6e-4 is the assumed optimal LR computed using a grid search reported in Liu et al. (2023). Subsequently, we pre-trained the same model using an LR of 6e-3, which is tenfold higher in magnitude than the former. […] Interestingly, we note that the training trajectory of LAWA approximates that of a model trained with a lower LR.“  - I don’t think I get what is the practical use-case of LAWA then if the vanilla model with tuned LR obtains the same result as reported in the Figure 2.
- “In practical LLM pre-training scenarios, where conducting a grid search is challenging due to the model’s size, adopting our proposed training recipe could be advantageous. One might select a higher LR (that doesn’t cause divergence) and train an LLM faster without compromising much generalization compared to conventional pre-training strategy.” - I buy this argument, however I am not sure how practical and significant this is given approaches like (Tensor Programs V, Yang et al, 2022) that are used to efficiently find hyper-parameters for large-scale experiments.

Marginal gains:
- Gains reported in Table 2 for Pythia checkpoints are imo not significant. After 105K or 141K steps, LAWA do not always perform better, and when it does the improvement is most of the times smaller than 1%, and no standard deviations are reported. In Figure 2 you report the case where the Vanilla model with tuned LR performs on par with LAWA. In Figure 1 you report gains at the end of the training, but it’s hard to assess if these are significant as you do not report standard deviation, also these gains are for smaller models (not LLMs).
- In Figures 4,5,7,8 the gains get smaller and smaller as you increase the numbers of training steps, and possibly anneal the LR. As mentioned earlier I don’t think that it’s fair to compare intermediate steps when the baseline’s training recipe wasn’t tuned for this number of steps. I think that this is possibly a major issue in the experimental setup. The same goes for Figure 6 where you compare the perplexity of intermediate checkpoint, while if the LR was annealed for the baseline for some smaller predefined number of steps the gains might have not be there. Furthermore, for 6.9B model which was trained for 33500 GPU hours, I don’t think that 1000-4000 GPU hours save is major, given extra complexity.

**Questions:**

A. In what cases is your method useful/significant given my concerns raised in the "Artificial setup" point?
B. I would ask the authors to reflect on my point where I argue that comparing intermediate checkpoints is not fair, and if possible to re-evaluate their experiments in a setup where they tune both the baseline and their method for some pre-specified compute budget.